# A semiconductor 96-microplate platform for electrical-imaging based high-throughput phenotypic screening

Shalaka Chitale [1], Wenxuan Wu [1,2], Avik Mukherjee [3], Herbert Lannon [1], Pooja Suresh[1], Ishan Nag[1], Christina M. Ambrosi[1], Rona S. Gertner[4], Hendrick Melo[1], Brendan Powers[1], Hollin Wilkins[1], Henry Hinton [2], Michael Cheah[1], Zachariah G. Boynton[1], Alexander Alexeyev[1], Duane Sword[1], Markus Basan [3], Hongkun Park[4,5] ✉, Donhee Ham [2] ✉ & Jeffrey Abbott[1,2,4,5] ✉

High-content imaging for compound and genetic profiling is popular for drug discovery but limited to endpoint images of fixed cells. Conversely, electronic-based devices offer label-free, live cell functional information but suffer from limited spatial resolution or throughput. Here, we introduce a semiconductor 96-microplate platform for high-resolution, real-time impedance imaging. Each well features 4096 electrodes at 25 μm spatial resolution and a miniaturized data interface allows 8× parallel plate operation (768 total wells) for increased throughput. Electric field impedance measurements capture >20 parameter images including cell barrier, attachment, flatness, and motility every 15 min during experiments. We apply this technology to characterize 16 cell types, from primary epithelial to suspension cells, and quantify heterogeneity in mixed co-cultures. Screening 904 compounds across 13 semiconductor microplates reveals 25 distinct responses, demonstrating the platform's potential for mechanism of action profiling. The scalability and translatability of this semiconductor platform expands high-throughput mechanism of action profiling and phenotypic drug discovery applications.

High-throughput screening is the dominant paradigm for profiling compounds based on biological activity, toxicity, and mechanism of action (MOA)[1]. One of the most informative screening tools is high-content imaging with feature extraction to create high-dimensional profiles (e.g., Cell Painting[2]). However, the technique only produces an end-point image of fluorescently labeled fixed cells, missing important characteristics of live cells and tissues. For example, barrier and water transport properties of epithelia are important for cancer[3], fibrosis[4], inflammation[5], and cystic diseases[6,7], yet they cannot be assessed using high-content imaging.

Impedance techniques can complement the shortcomings of imaging, providing live-cell morphology information—such as epithelial barrier properties—in real time, throughout the experiment. They are non-invasive and label-free[8], making them versatile for a wide range of biological questions without the need for fluorescent probes or cell line engineering. However, existing impedance devices, like transepithelial electrical resistance (TEER) assays[9] and other commercial devices (e.g., xCelligence RTCA[10] by Agilent Technologies, Inc., ECIS[11] by Applied Biophysics, Inc.), have drawbacks as they measure only one[10] or two[11] parameters per well using

[1]CytoTronics Inc., Boston, MA, USA. [2]John A. Paulson School of Engineering and Applied Sciences, Harvard University, Cambridge, MA, USA. [3]Department of System Biology, Harvard Medical School, Boston, MA, USA. [4]Department of Chemistry and Chemical Biology, Harvard University, Cambridge, MA, USA. [5]Department of Physics, Harvard Univerusity, Cambridge, MA, USA. ✉e-mail: hongkun_park@harvard.edu; donhee@seas.harvard.edu; jabbott@cytotronics.com

large electrode pairs, compromising both accuracy and interpretability of information content.

Consumer electronic complementary metal-oxide-semiconductor (CMOS) technology offers a potential solution to these challenges through integration of thousands of electrodes at cellular resolutions[12]. To date however, CMOS impedance-based measurements have been limited to single well prototypes. To fully harness the benefits of impedance techniques, they should be provided not only in high spatial resolution and thus high-accuracy, but also in a high-throughput form factor to enable the screening and profiling of compounds, and ultimately the development of novel and effective therapeutics.

In this work, we present an impedance platform for drug discovery applications that accomplishes both high spatial resolution and high throughput. By creating a semiconductor 96-microplate using custom designed CMOS integrated circuits (ICs), we achieve 4096 electrodes/well with a spatial resolution of 25 μm. To enable high throughput, miniaturized interfacing electronics enable readout directly inside standard cell culture incubators, accommodating up to 8× semiconductor 96-microplates per incubator. Beyond hardware specifications, our platform incorporates electric field-based measurements. These techniques capture >20 impedance parameter images per well at 5–15-min intervals, providing real-time and label-free insights into a wide range of live-cell morphological and functional properties.

The impact of our impedance platform spans various fields, including phenotypic discovery, toxicity and profiling assessments, and general live cell biology research. With its enhanced accuracy,

throughput, and multi-parametric information, our platform paves the way for transformative advancements in drug discovery and cellular analysis.

## Results

### Design and optimization of a 96 well CMOS plate, high throughput platform

To scale and improve spatial resolution and accuracy of impedance techniques for drug discovery applications, we custom designed a CMOS IC chip for integration into each well of a standard form factor 96-well plate (Fig. 1a–c). Each individual IC occupies an area of $18 \times 18$ mm$^2$ to support four wells, therefore a total of 24× ICs are mounted on a printed circuit board (PCB) to create the 96-microplate at a standard 9 mm well to well spacing (Supplementary Fig. 1). Electrical input/output signals are routed across the IC both horizontally and vertically to distribute electrical signals from connectors on the bottom side of the PCB across the plate (Supplementary Fig. 2). For readout and control, miniaturized interfacing data acquisition (DAQ) electronics create a universal serial bus (USB) interface to communicate with a computer.

A custom 96-well plastic (polyethylene terephthalate/PET) plate is attached via epoxy to the mounted CMOS ICs, designed with a standard form factor to be compatible with high-throughput instruments such as automated fluid handlers (Fig. 1b). Each well has a 140 μL maximum capacity and 120 μL working volume with a bottom diameter of 3.4 mm (Fig. 1c). A $64 \times 64$ array of 4096 pixels at a 25 μm pitch at the bottom of the well results in a $1.6 \times 1.6$ mm$^2$ total sensing area. Each pixel contains a gold electrode[13], a digital memory, and switches to

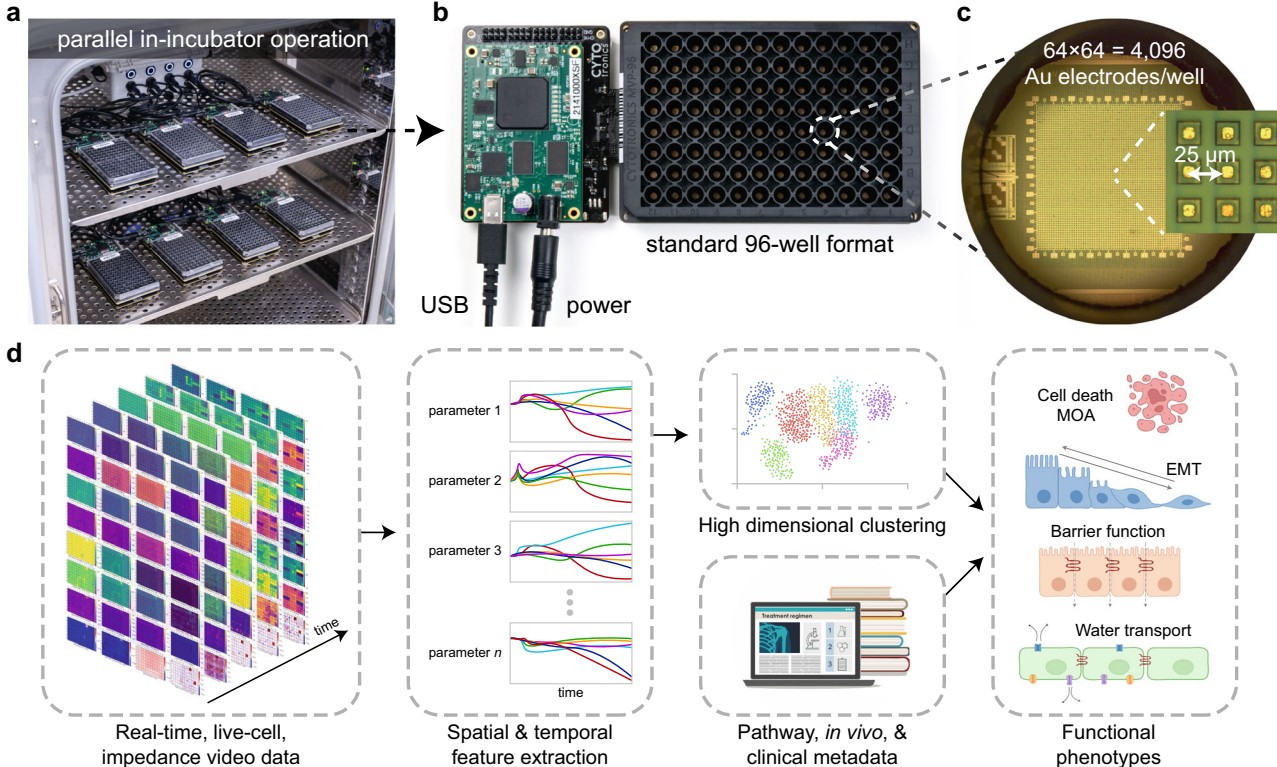

**Fig. 1 | A semiconductor 96-microplate platform for in-incubator, electrochemical high-throughput screening. a** Up to 8× complementary metal-oxide-semiconductor (CMOS) microplates operate in a cell-culture incubator for high-throughput electrochemical screening. The incubator regulates the ambient temperature, $CO_2$, and humidity. A universal serial bus (USB) and power hub connect the microplates to an external computer. **b** Each CMOS microplate contains 96 wells and connects to a miniaturized data acquisition system which forms a USB interface and provides power. The microplate is designed with standard

dimensions so that it is compatible with wet-lab automation (e.g. automated fluid pipettes) and other instruments (e.g. centrifuge, plate reader). **c** At the bottom of each well is a CMOS integrated circuit (IC) containing an array of $64 \times 64 = 4096$ Au electrodes, spaced at a pitch of 25 μm. Embedded circuits scan electrochemical measurements across the array generating impedance images every 5–15 min. **d** For high-throughput screening applications, the real-time, live-cell, impedance movies are fed through a data science pipeline and combined with known compound metadata to identify functional phenotypes.

form connections with peripheral circuits integrated on the CMOS IC (Supplementary Fig. 3). The pitch of the electrodes allows for near single-cell resolution, while the total sensing area enables measurements of entire cell sheets, useful for epithelial cell types and tissues.

The IC design and microplate construct represents a major advancement to state-of-the-art CMOS devices for biological measurement, where the large majority are formed in single well constructs[12]; Supplementary Table 1 provides a detailed comparison of our system to other impedance technologies. Some notable exceptions are Maxwell Biosystems AG[14] and 3Brain AG[15] which sell 6-well CMOS-based plates for neural applications. Our work is differentiated in two ways. First, by both forming the CMOS electronic 96-microplates and miniaturizing interfacing off-plate electronics, we can perform 8× plate operation (a total of 768 wells) directly in the incubator using simple power and USB hubs (Fig. 1a). In contrast, other CMOS or impedance microplates work off an instrument principle, where the plates plug into an instrumentation box. This limits measurements to a single plate per instrument[11,14,15], or up to 6 plates for a 2-electrode per well device[10]. In principle, our strategy allows scaling to even higher throughputs of 10 s or 100 s of plates through improved USB and power hub interfaces. Currently, we run 3 incubators and 24× CMOS microplates in parallel, with plans to expand in the near future. Second, we demonstrate impedance techniques capable of measuring a wide range of cell biology (Fig. 2). These cell-type agnostic capabilities open application areas beyond neuroscience, such as cancer and epithelial disease screening. Towards this end, the schematic in Fig. 1d summarizes the platform's utility for drug discovery applications via a cloud-based data science pipeline.

## Field-based impedance techniques measure unique and orthogonal cell properties

In addition to increasing CMOS-based measurement throughput for cell biology applications, we advance field-based impedance techniques[12] to measure unique live cell information. We use three different electric field configurations, two of which are named vertical field (VF) and lateral field (LF) like our previous work[12], but with many key improvements. Briefly, the VF configuration operates on the principle of biased nearest neighbor electrodes blocking lateral fields in solution, therefore causing the measured field to arrange vertically (Fig. 2a, top). Since our previous work, we've optimized the system to include multiple layers of blocking electrodes to properly arrange the vertical field. In contrast to the previous 3 × 3 group of electrodes (1 layer of blocking electrodes), we now use a 16 × 16 group of electrodes biased together and record the center 4 × 4 electrode group simultaneously (6 layers of blocking electrodes, Supplementary Fig. 4). Importantly, the parallel measurement of 4 × 4 = 16 center electrodes reduces the total scan time by 16×. To complement, the LF configuration is also modified to reduce scan time by measuring 16 electrodes simultaneously; each measured electrode is spaced at least 12 electrodes away from each other to minimize cross-electrode effects (Supplementary Fig. 5). A new field geometry is also measured, called the 'electrode impedance field', which uses the same pattern as the lateral field, but measures the current through the active electrode instead of the nearest neighbor readout electrode (Supplementary Fig. 5).

Another crucial advancement was to expand beyond using a single frequency of 5 kHz[12] to simultaneous measurement of four frequencies of 250 Hz, 1 kHz, 4 kHz, and 16 kHz. Impedance of the cell membrane is inversely proportional to the measurement frequency (the lipid bilayer acts as a capacitor), which results in distinct features of the cells measured at different frequencies[8,16]. Therefore, for each field measurement, four frequency signals are digitally added together and applied to the active electrode(s). A Fast-Fourier Transform (FFT) then calculates the four frequency magnitudes and phases and a Direct Current magnitude to create 9 impedance parameter maps—27 total

maps when measuring all 3 field configurations (Methods). The simultaneous multi-frequency approach is faster than sweeping the frequency to reduce scan time[16] and the upper frequency (16 kHz) is limited by our amplifier bandwidth. In comparison to other works which measure high frequency (>1 MHz) cross electrode capacitance[17] or paired electrode capacitance changes[18], our field-based impedance measurements distinguish multiple low-frequency tissue and cell parameters far away from the Debye capacitance sensing region, as discussed below. Higher frequency field measurements could accomplish similar capacitive sensing[17,18] to reveal different cell properties[8,16], but would require higher bandwidth op-amps at the tradeoff of more power consumption.

The three new field biasing schemes were enabled by the flexibility of pixel to peripheral circuit connections of the CMOS IC design (Supplementary Fig. 3) and were critical to enabling real-time measurement across the 96-microplate. The total scan time is a function of the lowest measured frequency (at least 4 ms per electrode for a 250 Hz cycle), total electrodes and number measured simultaneously (4096 electrodes total with 16 measured at a time), configuration programming time, and number of wells measured simultaneously (6 wells total)[16]. Each field geometry takes 40 s to scan the full plate; the 3 fields are typically performed in sequence resulting in 120 s (2 min) for a full scan of the 27 impedance parameters. For most cell types and experiments, we found that performing full scans every 15 min balances total data size (~17 GByte/72 h experiment) and time-course resolution. For scaling arrays to more electrodes while maintaining a similar scan time, the lowest frequency measured could be increased, or improved electronics could be implemented to scan more electrodes and/or wells simultaneously[16].

To connect the impedance measurements to cell morphology and function, and to demonstrate the orthogonality of the new field-based impedance parameters, we highlight an experiment in Fig. 2 using MDCK cells, an epithelial cell line that forms a strong cell barrier and has demonstrated apical to basal water transport resulting in tissue doming[19,20]. At lower frequencies (250 Hz, 1 kHz), the capacitive nature of the cell membrane's lipid bilayer results in a very high impedance—in the VF, this causes the field to pass through intercellular spaces resulting in measurement of the permeability of the tissue or barrier. In the MDCK experiment, an increasing VF 250 Hz magnitude is observed after confluency is reached reflecting the process of tight junction formation[21] and reaches a peak after the onset of water transport[22].

At higher frequencies (4 kHz, 16 kHz), the fields are sensitive to membranes in closer proximity to the electrodes. For the VF configuration, high frequency signals reflect cell flatness (or the inverse of height): the flatter the cell, the more cell membranes are near the electrode increasing the impedance. To this end, a spike in the VF 16 kHz magnitude is observed in the experiment shortly after plating, attributed to the suspended cells falling to the surface and spreading[23]. In the high frequency LF, cell substrate attachment is measured with high sensitivity: the closer the cells are to the electrode, the higher the LF 16 kHz magnitude. In the MDCK experiment, the cells reach confluence and attach strongly to the surface followed by a rapid detachment with the onset of water transport. The water transport process can be accelerated by media changes, as in Fig. 2b, and is also affected by plating density (Supplementary Fig. 6 and Supplementary Movie 1).

The different trends observed across the four examples in Fig. 2b are supportive that information measured at the different frequencies and fields is orthogonal in nature. In our previous work[12], we found that our old VF and LF configurations at a single frequency often resulted in similar impedance trends. The new biasing schemes and multi-frequency analysis greatly extends the ability to extract high dimensional morphology data in parallel.

Beyond magnitude trends of fields and frequencies, another capability of our system that is currently not available in other

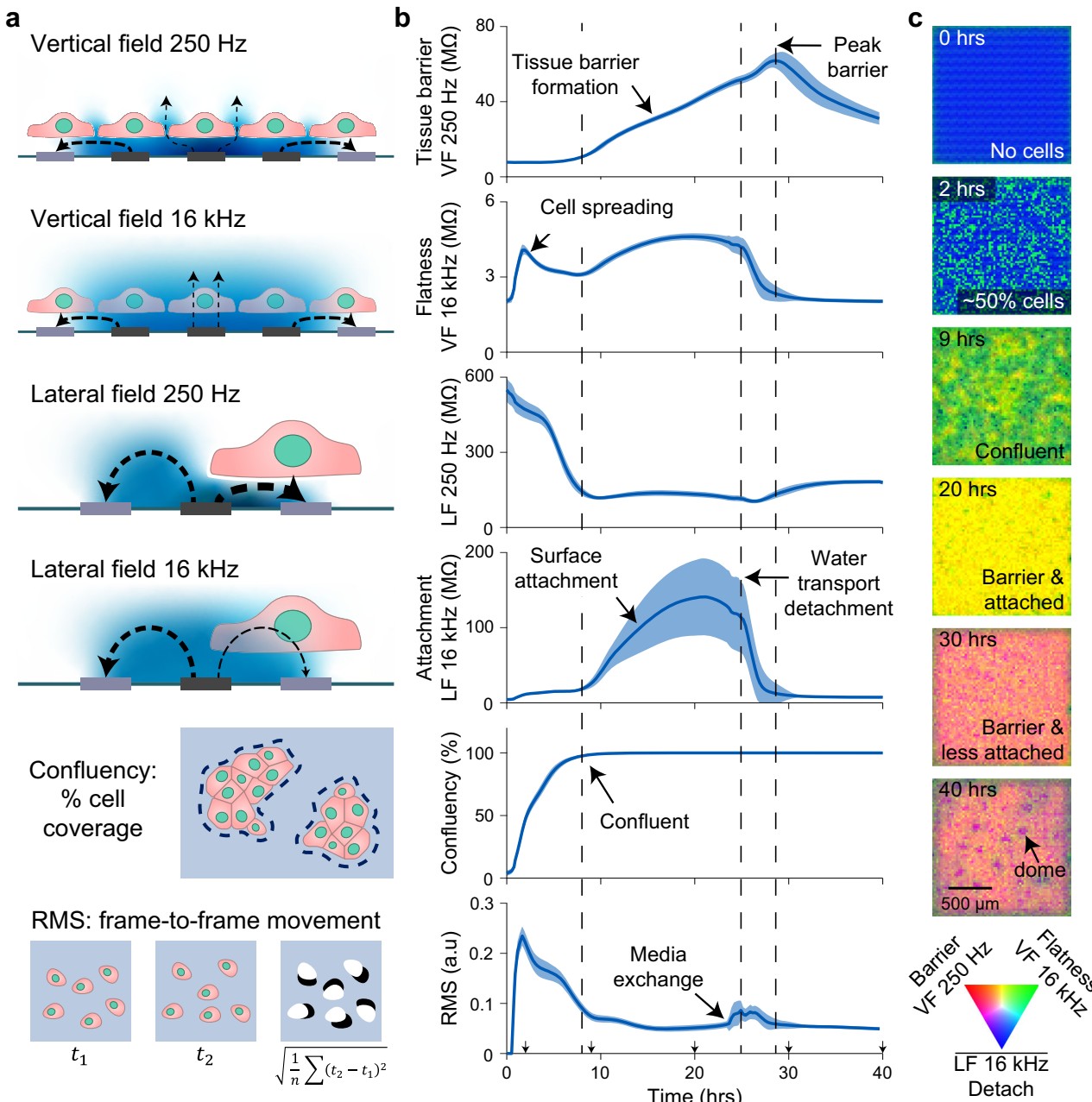

**Fig. 2 | Real-time, label-free field-based impedance measurements for live-cell functional morphology measurement. a** Schematics for the vertical field (VF) and lateral field (LF) configurations, confluency, and root mean square (RMS). Examples of measured electric field lines are shown between stimulation (dark gray) and return electrodes (light gray) and potential distribution in blue. At low-frequency (250 Hz), cell membranes block the electric fields whereas at higher frequency (16 kHz) the fields become more proximately sensitive. Confluency and RMS are calculated from the impedance images. **b** Parameter traces are shown for MDCK cells (canine kidney) plated on a 96-well CMOS microplate. Cells were seeded at time $t = 0$ hrs at a density of 40,000 cells/well and measured for 40 hrs. Media was changed at 24 hrs post seeding. The traces represent mean ± s.d. for 88 wells across the plate. The most representative biological parameter is labeled for each measurement, but a blend of biological parameters contributes to each measurement in total. Source data are provided as a source data file. **c** Impedance images of the MDCK cells from a single well at the indicated time points. The images were generated with VF 250 Hz in red, VF 16 kHz in green, and the inverse of the LF 16 kHz in blue.

impedance devices is the spatial resolution of our measurements. This generates electrical images of each well across each impedance parameters and can be analyzed similar to optical images.

Spatial features that can be extracted from the data include confluence, which is calculated as the percentage area covered by cells determined using a threshold (Fig. 2a, Methods). Cell locations are also used to mask impedance maps to account for differences in confluency and to omit data from non-covered electrodes—improving accuracy beyond traditional aggregate well techniques[9–11]. For

assessment of properties like cell-barrier, this extraction of data from only places with cells allows decreases of tissue barrier and cell-cell adhesion to be detected and differentiated from toxicity. In contrast, with single-electrode pair techniques[9–11], any decrease in confluency will shunt the measurement and result in a decreased barrier/impedance measurement.

Another unique parameter that is enabled in our system is the measurement of transient features like motility. The root mean square (RMS) of the difference from one measurement frame to another is

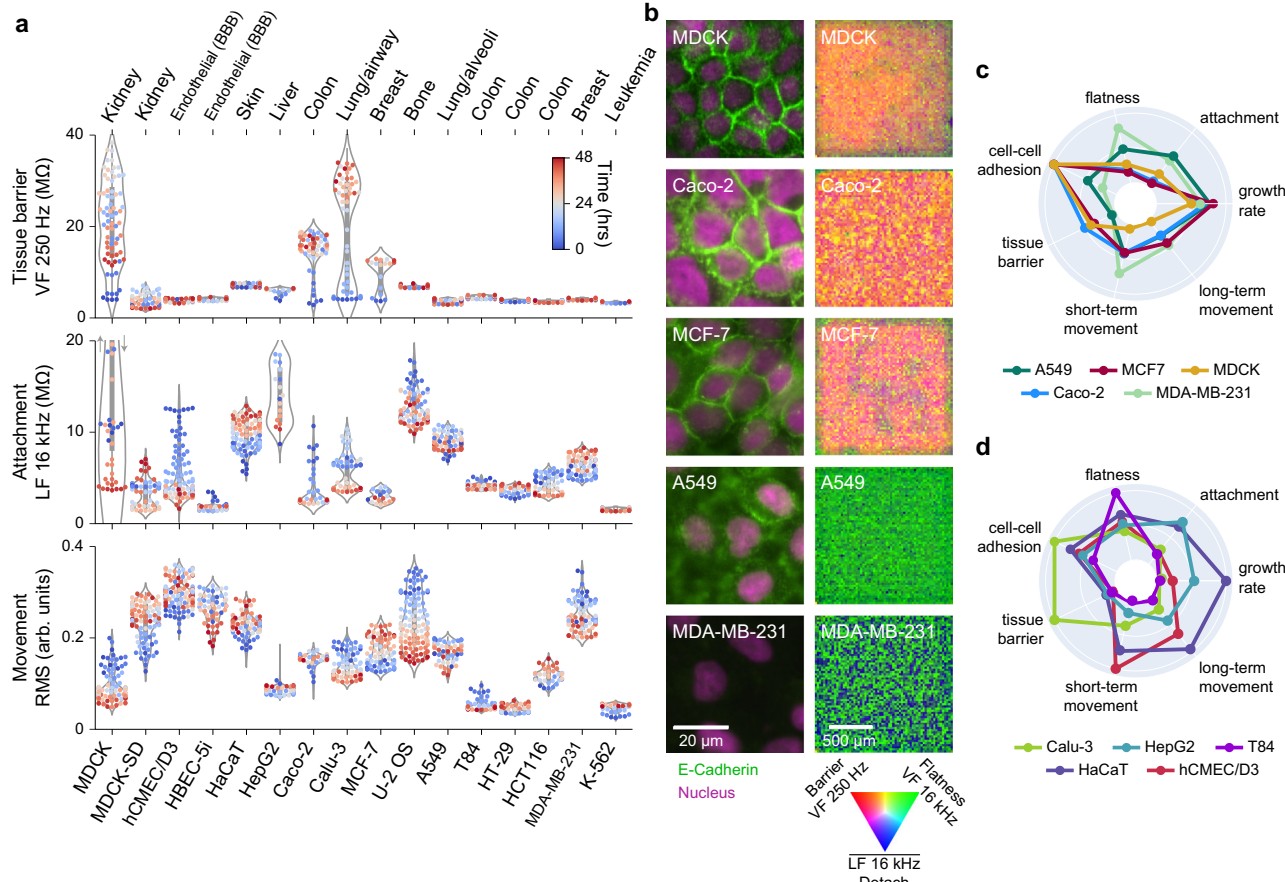

**Fig. 3 | Impedance measurements differentiate a wide variety of cell types and characterize cell state over time. a** Sixteen cell types were characterized using live-cell impedance measurements. Tissue origins are labeled; MDCK variants are canine, the remainder are human. Violin plots of three parameters show differences in tissue barrier (VF 250 Hz), cell-substrate attachment (LF 16 kHz), and movement/dynamic changes in morphology (RMS). Each data point represents a given measurement for a single well at a single timepoint from 0 h (blue) to 48 h (red). The shape of the violin plot represents the underlying distribution of datapoints over time. Real-time movies for all cell types at different seeding densities are shown in Supplementary Movie 1. **b** Immunofluorescence images (left) and impedance images (right) for various cell types show a range of epithelial tissue properties.

Cells were stained for E-Cadherin (green), a cell-cell adhesion protein, and for DAPI/nucleus (red). Immunofluorescence imaging was performed in 4 independent experiments, representative images from a single experiment are shown here. A representative well impedance image combines several impedance parameters into different colors (noted). Cells strongly expressing E-Cadherin at cell-cell interfaces (MDCK, Caco-2, MCF-7) show high tissue barrier in comparison to cells with dispersed E-Cadherin (A549) or no expression (MDA-MB-231). **c, d** Radar plots comparing the functional indices of the indicated cell lines across 7 cell lines characteristics; the radial axis is scaled from 0-1. Source data are provided as a source data file.

used for parameterizing transient features such as motility/movement and migration (Fig. 2a, Methods). Assessment of cell movement innately requires time and spatial resolution, therefore other techniques such as end-point fluorescent imaging or aggregate well impedance techniques cannot detect such live cell motility.

Lastly, spatial resolution enabled detection of tissue doming in MDCK cells (Fig. 2c)[19,20]; a transient and dynamic phenomenon in which transepithelial water transport leads to areas of increased water pressure under the cell sheet. This causes regions of the cell sheet to lift off in dome-shaped structures. In our measurements, we observe domes as circular decreases in attachment signals, represented in the bottom image of Fig. 2c as purple dots of ~150 μm diameter.

## Using impedance measurements to characterize a range of cell types

To test the sensitivity and versatility of our platform, we measured a wide range of well characterized cell types on our platform, spanning from primary epithelial cells to cancer epithelial cells, to suspension cells (Fig. 3). The cell types originate from the kidney, brain endothelial/blood brain barrier (BBB), skin, liver, colon, lung,

breast, and bone marrow/leukemia and are of human origin, except the MDCK cells which are canine origin. The impedance techniques measured characteristics of all tested cells (Fig. 3a); Supplementary Fig. 6 shows real-time traces for various plating densities (typically 10,000 to 40,000 cells/well) and Supplementary Movie 1 shows an example well for each density and 11 cell types for the first 48 h of culture growth.

The epithelial cells exhibited a strong attachment to the substrate and formed a continuous sheet with many tight cell-cell junctions that create a high barrier when confluent (Fig. 3a). Beyond MDCK, these cell types include Caco-2 (colon cancer cells that are used as a model of intestinal barrier and inflammation[24]), Calu-3 (lung adenocarcinoma cells used as a model of bronchial barrier[25]), and MCF-7 (luminal A type breast cancer cells exhibiting differentiated mammary epithelium properties[26,27]). In contrast, a variety of cell types from similar human tissue origins did not create barriers. These cell types include HepG2 (hepatocellular carcinoma), T84 (metastatic colon carcinoma), HaCat (immortalized human keratinocytes), A549 (lung adenocarcinoma cells derived from alveoli[28]), and MDA-MB-231 (triple negative breast cancer cells[29]). To complement the cancer cells, hCMEC/D3 (brain endothelial cells

used for modeling BBB function[30]) exhibited the highest motility of any cell type (see also Supplementary Movie 1).

Finally, K-562 (chronic myelogenous leukemia cells grown in suspension) were effectively measured and the presence of live cells above the electrodes identified. Traditionally, impedance measurements are only sensitive to adherent cells that are in direct contact with electrodes. However, our use of field-based techniques across the almost single-cell resolution array allows detection an estimated ~10–20 μm from the device surface[12]. As expected, suspension cells had the lowest cell attachment signal of all the tested cell types and did not form an epithelial sheet/connected tissue (Supplementary Movie 1).

Our platform's morphology measurements help identify important functional differences between cell types. For example, by using different low-frequencies in the VF, we can distinguish between tissue barrier and local adhesion at cell-cell junctions. We found that the VF 250 Hz measurement assesses the integrity of tissue barrier across an ~200 μm region, while the VF 1 kHz measurement focuses on local cell-cell adhesions. Immunofluorescence imaging for representative cell types helped form this correlation by detecting expression and localization of E-cadherin, a functional component of adherens junctions expressed in epithelia (Fig. 3b). Loss of E-cadherin can cause dedifferentiation and invasiveness in human carcinomas and is observed in cancer cells that have undergone epithelial to mesenchymal transition (EMT)[31]. We observed that cell types with high-levels of membrane E-cadherin and a regular epithelial morphology (MDCK, Caco-2, and MCF-7) showed a strong tissue barrier signal, displayed in the radar plot of Fig. 3c. In contrast, cells types like MDA-MB-231 which had undetectable levels of E-cadherin, did not exhibit signals for tissue barrier or local cell-cell adhesion. Interestingly, cell types such as A549, which lacked a tissue barrier signal, still displayed measurable cell-cell adhesion (Fig. 3c), indicating lower and more dispersed E-Cadherin levels (Fig. 3b). This correlation demonstrates how the multi-parametric, spatial approach of our platform provides a more comprehensive understanding of cell sheet properties compared to traditional techniques like the transwell assay[9], which only provides a global TEER readout. Furthermore, our other field-based impedance measurements assess unique morphological and dynamic properties such as cell flatness, surface attachment, short-term movement, long-term movement and growth rate, thereby enhancing our capability to distinguish functional differences (Fig. 3c).

To this end, other impedance methods have also reported differences between cell types based on impedance readouts[9,32–35]. For example, our observation that Calu-3 cells have a higher barrier than A549 cells was previously observed using a multi-well impedance device (Axion Biosystems)[33] and by conventional trans-well TEER measurements[35]. However, we additionally observe that A549 cells have a higher attachment and motility than Calu-3 cells, features that other impedance techniques cannot observe. Of note, the normalized barrier resistance taking into consideration the effective unit area of each electrode [$25 \times 25 \, \mu m^2$] for Calu-3 (~200 $\Omega \cdot cm^2$) and A549 (~20 $\Omega \cdot cm^2$) match value ranges from various literature sources using traditional TEER readouts[35], thus validating our measurement method.

To underscore the platform's capability in assessing live-cell function, we compared additional cell types from different tissues, each with distinct characteristics (Fig. 3d). For instance, hCMEC/D3 cells exhibit the highest short-term movement, HepG2 cells demonstrate high surface attachment, HaCat cells show the fastest growth rate and long-term movement, while T84 cells appear notably flat. To enhance this comprehensive functional assessment ability, we also demonstrated that our semiconductor 96-microplate is compatible with surface coatings such as collagen, fibronectin, and Poly-D-Lysine (PDL), commonly used to mimic extracellular matrix signaling of in vivo conditions[36]. Measurement of Caco-2 cells in wells with or without a collagen type I coating (Supplementary Fig. 7) revealed

similar growth and tissue barrier characteristics. Interestingly, a small difference in attachment was observed, reflecting the increased distance of the cells from the electrode due to the coating, further highlighting the sensitivity of our technique.

## Assessing culture heterogeneity in the context of epithelial-to-mesenchymal transition (EMT)

The high spatial and temporal resolutions of our platform can be used to study the heterogeneity of mixtures of epithelial and mesenchymal cell populations. EMT is a well-documented phenomenon associated with cancer and fibrosis, characterized by the progression of cancers to a more aggressive state and an increased probability of metastases[37]. During EMT, epithelial cells lose their barrier, undergo morphological changes, and acquire enhanced migration and invasion properties[38]. This transition occurs gradually, resulting in heterogenous cell populations. To model this process, we selected two breast cancer cell lines (Fig. 2a, b): MCF-7 shows an epithelial morphology and expression of E-cadherin while MDA-MB-231 shows a mesenchymal phenotype with no expression of E-Cadherin[29]. Of note, currently used impedance devices have been unable to distinguish between these cell types in head-to-head comparisons[32]. We plated MCF-7 and MDA-MB-231 cells at a low-plating density (10,000 total cells per well) and different ratios to measure differences in growth characteristics over 72 h (Fig. 4 and Methods).

Of the measurement parameters, we found that the RMS parameter, indicative of short-term movement, was able to best differentiate the cultures. RMS impedance images at 24 h show an intermediate range of cell movements for the 5000:5000 mixed population in comparison to the low movement from the pure MCF-7 (epithelial) and high movement of the pure MDA-MB-231 (mesenchymal) cultures (Fig. 4a). Over time, the distribution of the mixed population transitions from an almost 50/50 split between the pure distributions, to more MDA-MB-231 at 72 h (Fig. 4b). Quantification of the degree of similarity to pure population training distributions (Fig. 4c, Methods) shows the culture dynamics over time: for the 5000:5000 mix, an almost equal contribution of pure cell types (a 0.5 similarity score for both) transitions to an increasing MDA-MB-231 contribution (similarity score approaching 1) and decreasing MCF-7 contribution (similarity score approaching 0). This is attributed to the MDA-MB-231 cells having a faster growth rate as compared to MCF-7 cells within the co-culture. Co-cultures starting with fewer or more MDA-MB-231 cells show different starting contributions and extents of culture takeover; the model is also validated using pure test cultures (Fig. 3c, top and bottom).

Leveraging spatial information to assess population statistics over time can reveal subtle responses of culture sub-populations to stimuli and are more sensitive than cumulative, aggregate well readout approaches. For example, though previous literature could assess the 'energy' of a culture to compare cancerous versus non-cancerous cell types[32], they are not able to perform population statistics within the well itself to assess heterogeneity as there is no spatial information.

## Exploring the platform as a tool for drug discovery

Having established our platform's sensitivity to measure inherent differences in cell morphology and function, we performed compound screening across three distinct cell-types: MDCK (primary epithelial), A549 (cancer, epithelial-like), and MDA-MB-231 (cancer, mesenchymal). Our goal was to assess the effects of a common set of compounds targeting a variety of cellular processes such as cell division, DNA replication and inflammation to explore the range of functional phenotypes that can be observed (chosen compounds include Cytochalasin D, Vinblastine Sulfate, Paclitaxel, Alisertib, Bosutinib, Anisomycin, Dexamethasone, Getfitinib, Decitabine, Cyclophosphamide Monohydrate, and GSK 269962 A). Supplementary Figs. 8–10 show line traces of the measurements and

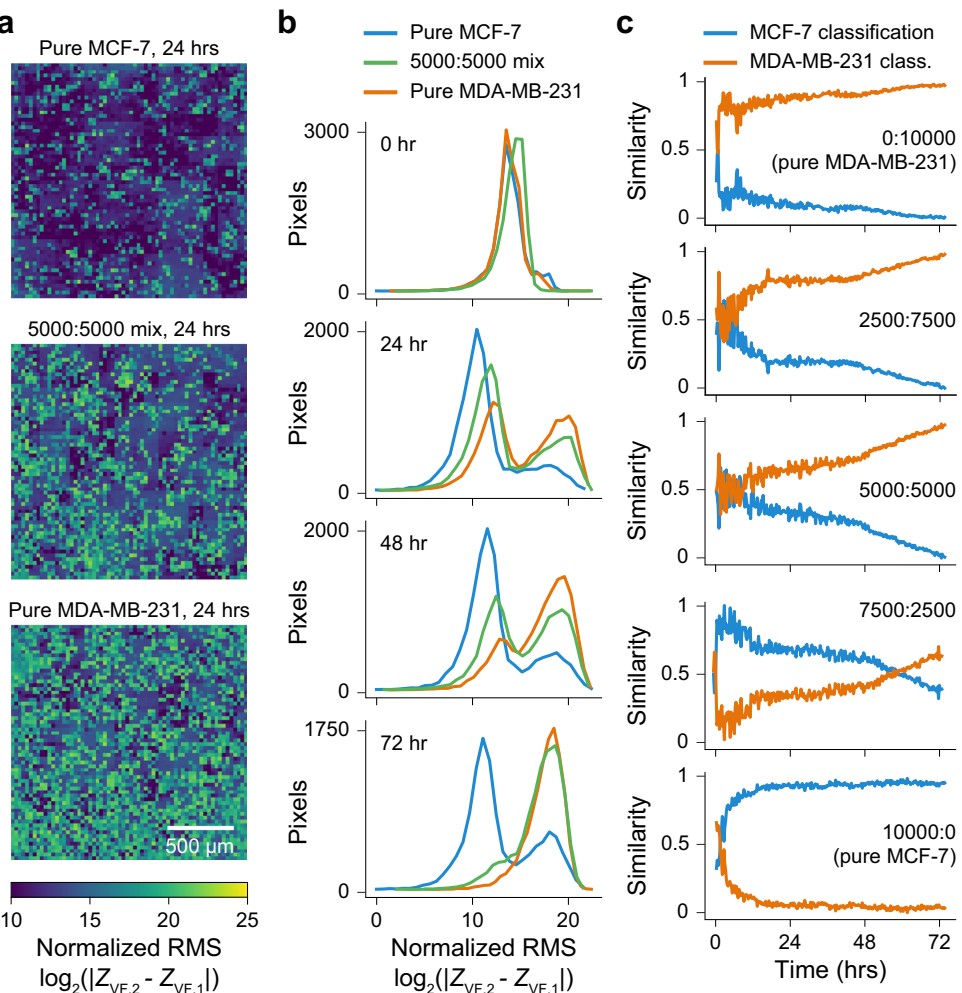

**Fig. 4 | Measurement of heterogeneity in mixed populations of epithelial and mesenchymal cell types. a** Normalized RMS impedance images for pure MCF-7 cells (epithelial phenotype), a 1:1 mixture of MCF-7:MDA-MB-231, and pure MDA-MB-231 cells (mesenchymal phenotype). Dark blue values of low RMS indicate low motility/movement, while green/yellow regions of high RMS indicate high motility/movement. **b** Distributions of normalized RMS across the 4096 pixels per well for MCF-7 cells (blue), 5000:5000 MCF-7:MDA-MB-231 (green) and MDA-MB-231 cells (orange) at time points from 0 to 72 h. The mixed population gradually transitions towards the pure MDA-MB-231, attributed to an increased growth rate of the MDA-MB-231 over the MCF-7 cells within the co-culture. **c** Graphs show the predicted percentage of MCF-7 (blue) and MDA-MB-231 (orange) cells over time for various co-culture ratios. The gray overlay indicates the model is not predictive until +5 h after plating (as determined by the pure population tests). Source data are provided as a source data file.

Supplementary Movies 2–4 provide well-plate movies of the pre- and post-compound effects over 72 h.

As highlighted in Fig. 2, MDCK cells are primary epithelial cells that form a strong barrier, exhibit a dynamic water transport phenomenon, and have low motility due to their formation of a tight epithelial sheet. Consistent with these inherent properties, we observed compounds that modulated barrier function (e.g., Cytochalasin D, Vinblastine Sulfate, Anisomycin) and water transport (e.g., Alisertib, Bosutinib) (Supplementary Fig. 8). These latter findings were particularly interesting as they correlated with in vivo results for autosomal dominant polycystic kidney disease (ADPKD). In humans, ADPKD involves cell overgrowth and a dysregulation of water transport function, resulting in fluid accumulation in cysts[39]. In our MDCK drug treatments, we found that Alisertib, an Aurora kinase inhibitor, enhanced water transport as evidenced by a sharp decrease in cell surface attachment and increased doming (Fig. 5 and Supplementary Movie 2). In contrast, Bosutinib, a multi-kinase inhibitor, hindered water transport as indicated by slower detachment of cells and the absence of domes. Interestingly, Alisertib has been shown to exacerbate ADPKD in animal models[40] while

Bosutinib was in Phase II clinical trials for treatment of ADPKD[41]. These in vivo findings align with our observed differential signatures in MDCK cells, underscoring the therapeutic implications of our innovative water transport measurement technique.

A549 cells demonstrated the ability to differentiate MOAs for drugs with similar outcomes[42]. For example, Paclitaxel and Vinblastine, which caused similar amounts of cell death, exhibited distinct dynamic changes in other morphological properties, such as surface attachment (Supplementary Fig. 9, Supplementary Results). This capacity to differentiate MOA using multiple measured parameters suggested the potential of our platform in large-scale phenotypic profiling for drug discovery. Additionally, our study emphasized the significance of selecting the appropriate cell type when screening compounds. For instance, Dexamethasone, an anti-inflammatory compound that has been studied in the context of EMT, caused a decrease in motility in the mesenchymal MDA-MB-231 cell type[43], while it increased cell-cell adhesion in the epithelial-like A549 cells[44] (Supplementary Figs. 9, 10 and Supplementary Results). These results stress the importance of choosing the most suitable cell-type to capture the range of phenotypes relevant to the disease or biological question of interest.

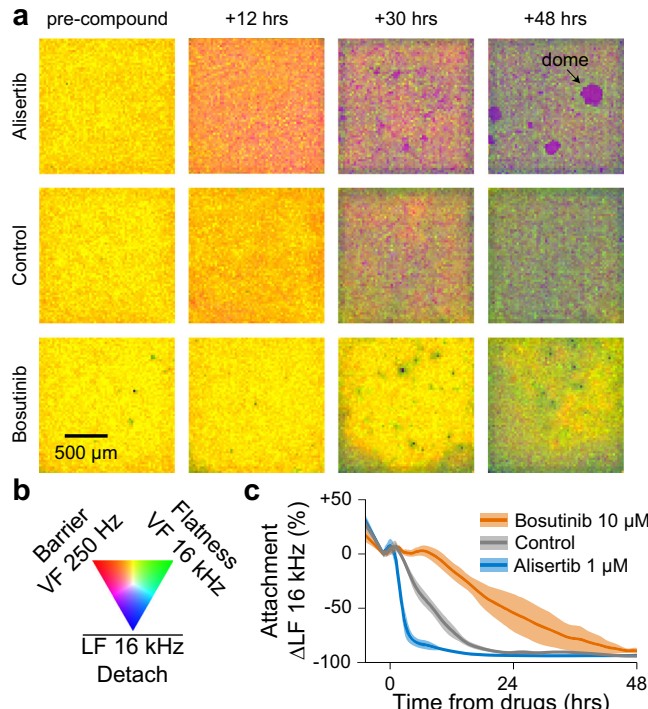

**b**
Barrier
VF 250 Hz

Flatness
VF 16 kHz

LF 16 kHz
Detach

**c**

Bosutinib 10 μM
Control
Alisertib 1 μM

Attachment ΔLF 16 kHz (%)

Time from drugs (hrs)

**Fig. 5 | Water transport phenotype for ADPKD drug discovery applications.**
**a, b** A set of 10 compounds were tested against the water transport phenotype exhibited in MDCK (canine kidney) cells; Supplementary Movie 2 shows the full 96-well CMOS microplate movie. Select time point electrochemical images for 1 of 3 replicates for Alisertib (1 μM) and Bosutinib (10 μM) are shown in (**a**) with a color-map in (**b**). Domes were only observed for tissues treated with Alisertib. **c** Line traces show Alisertib accelerated and Bosutinib delayed the water transport detachment (LF 16 kHz) with respect to the control; mean ± s.d. for 3 wells per condition. The differential results match animal study results (Main text) for autosomal dominant polycystic kidney disease (ADPKD), suggesting the water transport phenotype could be predictive of ADPKD efficacy. Source data are provided as a source data file.

## Compound profiling and mechanism of action (MOA) identification using high-dimensional impedance measurements

To expand our study and explore the full capabilities of our platform, we performed high-dimensional phenotypic profiling of compound responses using live-cells. For the proof of concept, we chose to screen against A549 cells for their intermediate values across many morphological parameters (Fig. 3c) and large dynamic range of compound responses[42] (Supplementary Fig. 9). The screening involved 904 compounds applied across 13 semiconductor 96-microplates, including FDA approved compounds and a subset from a targeted diversity library. To ensure robustness, we incorporated five positive controls onto each plate.

Compound activity was first gauged using a phenoactivity score[42], calculated as the overall response across all parameters from the DMSO negative control (Methods). Of the 904 compounds applied, 600 had phenoactivity that were significantly above the variability of the range of DMSO responses (outside of the DMSO mean ± 2σ). Compounds with low phenoactivity (within the DMSO mean ± 2σ) were labeled as 'no response' due to their overlap with the negative control (Fig. 6a). A histogram of the phenoactivities in Fig. 6b shows the DMSO cutoff; of note, the positive controls exhibited a range of phenoactivities from subtle to highly distinct.

To evaluate the reproducibility and robustness of identified phenotypic signatures, we then performed a principal component analysis (PCA) to assess measured response variance (Methods). The unbiased PCA generates a space that maximizes variance—the co-location within

the PCA space of the positive control replicates emphasizes the robustness of our impedance-based data acquisition approach (Supplementary Fig. 11). Furthermore, we explored the use of a linear discriminant analysis (LDA) model to determine if the positive controls could be clustered by their measurement signatures. The positive control data was split into training and test data sets (Methods); after training, the model assigned clusters with a precision score of 1, confirming distinct and separable signatures (Fig. 6c).

A similar approach was then adopted to assess the more diverse effects of the compound library. Unbiased clustering using the PCA identified compounds with similar effects on cell state. Notably, many clusters exhibited similar target and pathway annotations, indicating the ability to relate functional changes to specific targets or MOAs. The clusters were then used to train an LDA model (Fig. 6a), resulting in the differentiation of 25 distinct compound response clusters, including 1 no response/DMSO cluster and 5 positive control clusters. The model was then validated in two ways. First, we tested that randomly assigned labels did not form distinct clusters using the same LDA model parameters, obviating the curse of dimensionality[1] (Supplementary Fig. 12). Second, we divided the data set into a training and test set. Compounds were assigned to the correct cluster with a precision score of 0.68, which reflects a high accuracy for single replicates of diverse compounds (Supplementary Fig. 13).

All the compounds screened are annotated with well-characterized bioactivity (details in Methods). Thus, we were able to retroactively use the known bioactivity to examine functions that clustered together based on their effects on cells. In some cases, compounds with the same annotated MOA cluster together, pointing to a similar effect on cell state. However, compounds with different MOA may also cluster together, based on their effect on related functional/morphological pathways. To gain deeper insights into the separation of compounds into specific clusters and their effects on cell state, we examined time traces for selected clusters (Fig. 6d). The traces showed orthogonal measurements across the morphological parameters, differing in both magnitude and behavior over time. To facilitate comparison, we summarized the time traces using quantitative metrics termed bio-basis (Fig. 6e, details in Methods). Radar plots revealed the strength of our technology in classifying compounds causing cell death. Of the 6 clusters highlighted, 4 consist of compounds that affect cell proliferation or cause cell death through inhibiting DNA synthesis or cell cycle (clusters 1, 2, 3 and 5). In addition to the effects on cell confluence, each cluster exhibited distinct morphological effects and therefore varied bio-basis specific to their respective MOA. Compounds targeting microtubules (cytoskeletal signaling) or causing G2/M arrest (cluster 1, Supplementary Table 2) affect cell shape as measured by flatness. DNA synthesis inhibitors (nucleotide analogs) (cluster 2) primarily affect cell attachment and dynamicity. Compounds that either induce or inhibit the repair of double stranded DNA breaks (cluster 3), one of the most lethal types of DNA lesions, produce the highest cell death rate and a significant loss in attachment during cell death. CDK inhibitors (cluster 5) affect barrier strength, cell flatness and staticity (reduced cell movement). Cluster 6, comprised of various kinase inhibitors, causes a rapid and drastic loss in barrier, attachment, and movement. These functional insights provide new information on effects of known compounds and can help link unknown compounds to specific cellular pathways. Additionally, we identified a cluster primarily composed of GPCRs involved in neuronal signaling (cluster 4), which displayed a subtle yet rapid detachment response. This type of transient response can only be captured by our platform's high spatial and temporal resolution across parameters.

In conclusion, we demonstrate our platform's immense potential for compound profiling and broad applications in phenotypic assays for drug discovery. The high-dimensional data generated, coupled with the spatial and temporal resolution, enables the characterization

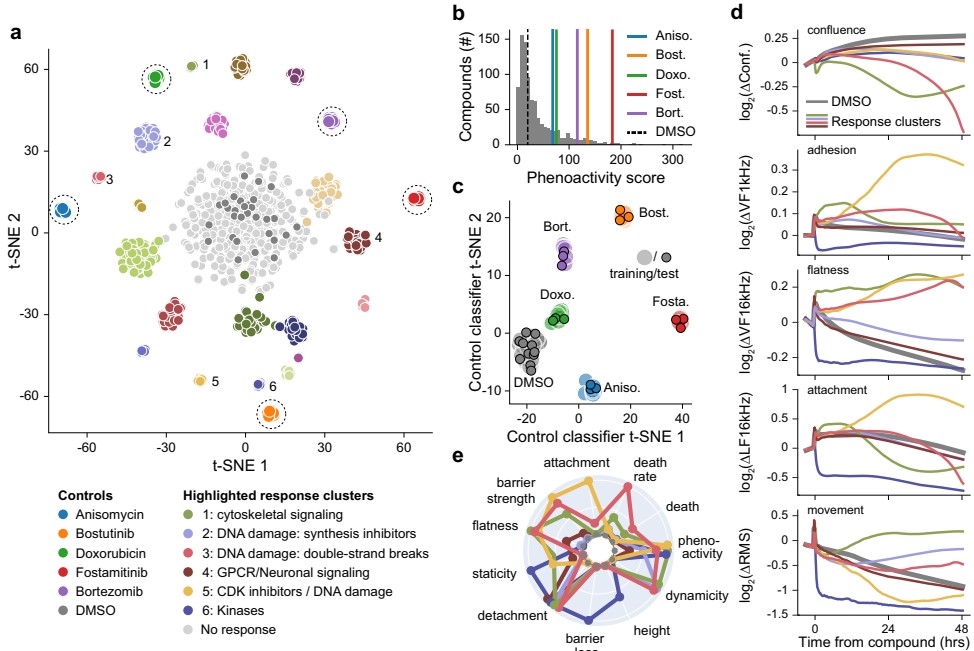

**Fig. 6 | A proof-of-concept screen of 904 compounds using 13× semiconductor 96-microplates to perform high-dimensional profiling and MOA identification.** **a** A549 (lung/alveoli) cells were treated with a library of 904 compounds at a concentration of 10 μM. A linear discriminate analysis (LDA) and a t-distributed stochastic neighbor embedding (t-SNE) model identified 25 distinct clusters using the high-dimensional impedance measurements. The legend indicates key high-lighted clusters (numbered from 1–6) and positive control compounds (indicated with dashed circles). **b** Compound phenoactivity scores characterize the extent of measured activity; 600 compounds exhibited phenoactivity separated from the DMSO negative controls, indicated by dotted line. Positive controls are displayed using colored vertical lines for reference. **c** A t-SNE plot showing the output of training and test data of a LDA based model for classifying control compounds. The 13 plate replicate data set was divided into a training set (filled circles) and test set (outlined circles). The test set was classified with a precision score of 1, indicating distinct and separable responses. **d** Time traces for the indicated measurements for numbered clusters from (**a**) across selected impedance measurement parameters. **e** Radar plots comparing 11 bio-basis for numbered clusters from (**a**) the radial axis is scaled from 0-1. Source data are provided as a source data file.

of compound responses and the identification of distinct phenotypic signatures associated with specific targets or MOA.

## Discussion

In this work, we describe the design and implementation of a custom semiconductor 96-microplate device with the potential to scale to a high-throughput screening system. We demonstrate its ability to measure a wide range of cell types using impedance measurement techniques and accurately distinguish between an array of biological responses to compounds. Furthermore, we perform a proof-of-concept screen to demonstrate its utility in drug discovery and the power of high dimensional data in identifying and profiling compound MOA. Altogether, our work significantly enhances the capabilities of impedance devices in drug discovery and cell biology research.

Scalability and versatility are crucial attributes for a technology employed in drug discovery. Our miniaturized data acquisition system allows parallel operation of multiple plates directly in the incubator, enabling unlimited scalability. In contrast, other impedance devices utilize an instrument design where plates plug into a box for measurement, reducing scalability. For eventual utility in drug discovery where 10,000 s of compounds are routinely screened, additional miniaturization of our 96-microplate to more dense form factors (e.g., 384 wells per plate) and/or more plates per incubator setup will be required. Major considerations to achieve this scale include management of high-speed data interfaces, thermal considerations with electronic power, and data processing pipelines.

In addition to scale, our measurement techniques reveal unique insights into cell function for versatile cell biology applications. Currently, CMOS devices are used mostly for neural applications, while most macroscale electrode impedance devices are only able to measure a bulk impedance signal per well and lack the ability to measure

orthogonal parameters in diverse cell types. Multiple biological processes such as cell death, loss of barrier, and change in attachment of cells are all reflected in the change in impedance per well and cannot be deconvolved. By combining field geometries and frequencies, we can measure distinct independent properties such as cell death, barrier, attachment, flatness, and motility. With the capability to measure over 20 morphological properties of live cells, our platform can build an information rich understanding of cell state.

Comparing our platform to other methods, high-content imaging with feature extraction (e.g., Cell Painting[2]), is the only other cell-based measurement approach capable of providing similar data dimensionality. However, the speed of image acquisition and limitations in phototoxicity/bleaching make it unsuitable for capturing dynamic morphological changes of live cells. Further, the properties of cells that can be acquired are constrained by the organelles that can be fluorescently stained and cell types are often chosen for ease of imaging (e.g. U-2 OS). By contrast, our platform can measure any cell type, label-free. Thus, our device can bring high-dimensionality assessment to a wide variety of disease and cell biology areas.

Data robustness and reproducibility are crucial for high-throughput drug discovery applications. Real time monitoring allows us to assess cell state before compound addition, which is not possible with end-point assays such as high-content imaging which requires cell fixing. By ensuring a similar starting cell state and monitoring of positive controls, we reduce the overall variability in data. For example, pre-compound summary statistics for the 13-plate proof-of-concept screen of Fig. 6 are compared in Supplementary Fig. 14, and unbiased and trained positive control replicate analysis are in Supplementary Fig. 11 and Fig. 6c. Outlier wells are identified even before compound addition using a cell-readiness assessment dashboard in our data science environment.

For novel drug discovery applications, our platform is the only technology offering a functional readout for water transport. The strong correlation observed between compound responses and water transport phenotypes of MDCK cells (Fig. 5 and Supplementary Movie 2) to animal and human studies of ADPKD, suggests potential applications for ADPKD screening[7]. Moreover, water transport-related phenotypes in other cell types, such as Caco-2 (as demonstrated in Supplementary Fig. 15 with drug-induced doming) and Calu-3, could serve as effective models for diseases like chronic diarrhea or cystic fibrosis. Furthermore, our platform's ability to separate toxicity, tissue barrier, and cell-cell adhesion properties could find strong utility in screening for compounds that enhance barrier tightness in context of gut diseases such as inflammatory bowel disease (IBD) or Crohn's disease[45,46].

The real-time, functional readouts provided by our platform enable the observation of cell state changes over time, providing valuable information on the intermediate and transient cell states in drug responses. This capability is particularly useful for determining cell death MOA: as cell death is innately a physical process, our platform is highly sensitive to the morphology of cell death and its 'order of operations'[47]. The ability to profile MOA in a label-free manner could find extensive use in in vitro toxicology, which often relies on end-point readouts that are blind to cell state transitions and potential off-target activities[48]. Identifying compound MOAs at the initial stage of high-throughput screening will dramatically reduce the time to lead selection and optimization, and ultimately make pre-clinical drug discovery more efficient and predictive of in vivo effects.

## Methods

### Cell culture and seeding
All cell lines were obtained from ATCC (unless otherwise noted) and maintained in a humidified incubator at 37 C and 5% $CO_2$. MCF-7 (HTB-22), A549 (CCL-185), MDA-MB-231 (HTB-26), MDCK (CCL-34), Calu-3 (HTB-55), U-2 OS (HTB-96), T84 (CCL-248), HaCaT (T0020001, AddexBio), and HCT116 (CCL-247) cells were cultured in DMEM (10017CV, Corning) supplemented with 10% FBS (35011CV, Corning). Caco-2 (HTB-37) cells were cultured in EMEM (10010CV, Corning) supplemented with 20% FBS. HepG2 (HB-8065) cells were cultured in EMEM supplemented with 10% FBS. K-562 (CCL-243) cells were cultured in IMDM (12440053, Gibco) supplemented with 10% FBS. HT-29 (HTB-38) cells were cultured in McCoy's 5 A medium (1660082, Gibco) supplemented with 10% FBS. hCMEC/D3 (SCC066, Sigma Aldrich) and HBEC-5i (CRL-3245) cells were cultured in EndoGRO (SCME004, Sigma Aldrich).

Cell lines were seeded in the semiconductor 96-microplates at various densities as indicated. Plates were coated with Collagen I from rat tail (354236, Corning) according to the manufacturer's instructions. All measurements, both pre- and post-compound treatment were performed in an incubator regulating $CO_2$, humidity, and temperature.

### Immunofluorescence imaging
For immunofluorescence imaging of Fig. 3, cells were grown on glass coverslips and allowed to grow for 48 h. Cells were then fixed in 4% paraformaldehyde (19943.K2, Thermo Fisher), washed, permeabilized with 0.5% Triton-X 100 (A16046.AP, Thermo Fisher) in PBS (21040CV, Corning), and then subjected to antibody incubation. The antibodies used was anti-E-Cadherin (3195 T, Cell Signaling Technology, 1:500) and Goat anti-Rabbit (H + L) Highly Cross-Adsorbed Secondary Antibody, Alexa Fluor Plus 488 (A32731TR, Invitrogen, 1:1000). Coverslips were mounted in Vectashield antifade mounting medium with DAPI (H-1800, Vector Laboratories).

Imaging was performed using a Yokogawa W1 spinning disk confocal on an inverted Nikon Ti fluorescence microscope equipped with Hamamatsu ORCA-Fusion BT CMOS camera (6.5 μm² photodiode), Lumencor SOLA fluorescence light source, and Nikon LUN-F XL

solid state laser combiner: 405 nm (80 mW), 445 nm (35 mW), 488 nm (80 mW), 514 nm (50 mW), 561 nm (65 mW), 640 nm (60 mW). Imaging was done using the widefield modality of the microscope with a 20X dry objective.

### Impedance measurements
To accomplish the field-based impedance measurements across multiple frequencies, we apply a voltage stimulation which is the summation of the four different frequency signals and measure return currents using a transimpedance amplifier (TIA) with a feedback gain ($R_2$ in Supplementary Fig. 3) of 18 MΩ. The magnitudes of the AC voltage signals are scaled to create similar output amplitudes that are measured by the TIA (0.2 V/250 Hz, 0.08 V/1 kHz, 0.04 V/4 kHz, and 0.02 V/16 kHz for lateral field, 0.25 V/250 Hz, 0.1 V/1 kHz, 0.04 V/4 kHz, and 0.025 V/16 kHz for vertical field). Six of the 96 wells are scanned at a time taking 2.5 s, resulting in a total scan time across the well plate of 40 s. A fast Fourier transform (FFT) is used to extract the magnitude and phase of each of the four frequencies and the DC component—9 impedance parameter maps for each field configuration. We observe that different frequencies contain different types of biological information (highlighted in Fig. 2) but are still biologically understanding the extant of information captured across the magnitude, phase, and DC information.

### Cell location masking, confluence, and root-mean-square (RMS) calculation
A reference impedance map is used to determine the location of cells by setting a threshold above the electrodes' default impedance in solution. The presence of a cell above an electrode will then increase the impedance beyond the threshold for detection. The mapping of electrodes occupied by cells is referred to as the cell mask. Typically, the VF 4 kHz map provides the best contrast for generating the cell mask. The cell mask is then used to calculate confluence as a percentage of electrodes occupied by cells for a given well. For magnitude, phase, and DC measurements, the median value of pixels with cells is then calculated. Additional metrics of the well distributions are being explored (e.g. standard deviation, 10%/90% distribution markers) but are not reported in this work. An additional epoxy mask is calculated in a similar manner as the cell mask to remove pixels which have spillover epoxy from the plastic well plate attachment, as seen in Supplementary Movie 4 as yellow regions. Before each cell plating, a reference measurement is taken in empty culture media to calculate the cell mask and epoxy mask.

Transient features such as motility and migration are generated from the impedance movies through a root-mean-square (RMS) calculation. For these calculations, only the epoxy mask is applied. The difference between two image frames is taken, then the RMS calculation is performed across the pixels of the difference map. To normalize for changes in magnitude, the calculated RMS is then divided by the median value of the cell-masked well distribution. The VF 4 kHz map is used for the RMS parameter generation throughout this paper.

### Statistics and reproducibility
All measurements done on the device include at least three technical replicates on each plate, unless otherwise noted as in the drug screen. Data from wells that were electronically faulty was excluded from the analysis. Parameters for exclusion were predefined. Electronically faulty wells resulted in no signal/ saturating signal. No statistical method was used to predetermine sample size. The investigators were not blinded to allocation during experiments and outcome assessment.

### Population statistics calculation
For the experiment in Fig. 4, MCF-7 and MDA-MB-231 cells were co-cultured in the same wells at different starting densities (MCF-7/MDA-

MB-231) of 2500/7500, 5000/5000, 7500/2500, 0/10,000, and 10,000/0 cells per well, respectively. The pure populations of either cell type were used as controls for the experiment. Empirical distributions of each of the experimental conditions were created by randomly selecting 3 wells for each of the starting conditions for the RMS (short-term movement) measurement. Reference distributions were generated by selecting 3 additional wells for each of the pure cell cultures. Wasserstein distance (earth mover's distance) was calculated, at each time point for each experimental distribution to both the reference distributions. The complement of the ratio of the two distances was then computed as a measure of the relative distance of the experimental distribution to the reference distributions.

### Compound treatments
Compound treatments were performed using a half media exchange. Compounds were prepared at 2X concentration in cell culture media, and the 2X compound solution was then added to plastic 96-well plates. Compound dilutions were performed with a constant DMSO (D2650, Sigma Aldrich) concentration and DMSO concentration per well was <1% (v/v) for all experiments. 2X compound in media was temperature and $CO_2$ equilibrated prior to addition. Before compound addition, half the media from each well of 96-well plate was removed.

Compounds used in this study were obtained from Selleckchem: Alisertib (S1133), Anisomycin (S7409), Bortezomib (S1013), Bosutinib (S1014), Cyclophosphamide monohydrate (S2057), Cytochalasin-D (S8184), Decitabine (S1200), Dexamethasone (S1322), Doxorubicin (E2516), Fostamatinib (S2625), Gefitinib (S1025), GSK 269962 A (S7687), Paclitaxel (S1150), and Vinblastine Sulfate (S4505).

### Compound libraries
All compounds were obtained from commercially available sources. A subset of 341 FDA approved compounds were selected from the FDA-approved Drug Library (L1300, Selleckchem), and a subset of 563 compounds were selected from the Targeted Diversity Library (HY-L099, MedChemExpress). Both libraries cover a range of targets and bioactivities. Bioactivity information for each compound has been previously reported in literature.

### Proof-of-concept screen and data-analysis
For the compound profiling screen in Fig. 6, A549 cells were seeded at a density of 15,000 cells/well across 13 semiconductor 96-microplates. Cells were allowed to attach and grow for 24 h. Single replicates of each compound at a concentration of 10 μM were then added at 24 h using the half media method as described in the Methods section. Compounds were distributed randomly across plates and not grouped by MOA. Endplate edge wells (columns 1 and 12) were excluded, and positive and negative controls were included on the interior of each plate. 13 plates in total were needed for the screen. The full set of impedance measurements were taken every 15 min.

Five positive controls were included on each plate: Anisomycin (100 nM), Bosutinib (10 μM), Doxorubicin (1 μM), Fostamatinib (1 μM) and Bortezomib (1 μM). 0.1% (v/v) DMSO was used as a negative control.

### High-dimensional data analysis
All measurement parameters were normalized to a timepoint 1 h before compound addition to calculate relative changes. The array of normalized parameters at timepoints logarithmically spaced from compound addition to +48 h were used for a principal component analysis (PCA). We found the logarithmic time spacing balanced rapid binding effects and longer-term effects increasing/decreasing over the full 48 h of compound treatment. Unbiased agglomerative clustering was then performed using the top 20 PCA dimensions. The cluster assignments produced by the clustering algorithm were used to label the dataset for the subsequent model described below.

The time-normalized measurement parameters for all wells in the screen were labeled with cluster assignments generated by the cluster algorithm. This dataset was then fed into a linear discriminant analysis (LDA) model for supervised classification. Positive controls and 19 compound clusters separated into clearly distinguishable clusters. The LDA model was validated by generating a second dataset with randomized cluster assignments. Performing the LDA model, maintaining the same model parameters, on the second generated data set produced overlapping indistinguishable clusters (Supplementary Fig. 12). To evaluate the model's performance and the robustness of the identified clusters, we split the cluster-labeled dataset into a training and test sets with a 80/20 split. After training the model with the training set, predictions were made on the test set with an accuracy score of 0.68 (Supplementary Fig. 13). Similarly, we also evaluated the robustness of the positive control responses using training and test sets across the 13 plates and obtained an accuracy score of 1.

### Cell line functional index calculation
We calculated functional indices across 7 different characteristics: cell flatness, tissue barrier, cell-cell adhesion, attachment, growth rate, short-term movement and long-term movement. Growth rate was calculated by taking the slope of the confluence over time within the first 12 h after cell seeding. The remaining 6 indices took the steady-state values between 36 and 48 h after cell seeding across their respective measurements. Each functional index was scaled from 0 to 1 using the minimum and maximum values measured across the 16 tested cell types. Thus, a functional index of 0.5 means that the particular cell type has a median value, and an index of 1 means that it has the highest value of the cell types tested.

### Bio-basis calculations for compounds responses
We calculate 11 bio-basis from 5 measurements parameters. Eight of the 10 bio-basis were calculated using the area under the curve of a given measurement over time between the compound and DMSO response. All measurement parameters were normalized to a timepoint 1 h before compound addition to calculate relative changes. The extent of a measurement increase (in the case of attachment, barrier strength, dynamicity/increased movement and cell flatness) was calculated by taking the positive area under the curve where the compound response increased relative to DMSO. On the other hand, the extent of a measurement decrease (in the case of detachment, barrier loss, staticity/decreased movement and cell height) was calculated by taking the negative area under the curve where the compound response decreased relative to DMSO. Cell death was calculated by taking the largest decrease in confluence relative to DMSO. Death rate was calculated by determining the most negative slope in confluence over time. In addition to the 10 bio-basis, we developed a phenoactivity score, which measures the cumulative magnitude of the compound effect across all measurement parameters by a modified residual sum of squares between a compound and the DMSO response. Compounds with a high phenoactivity have a large response magnitude compared to DMSO across multiple parameters, while those with a low phenoactivity implies a more subtle effect.

### Reporting summary
Further information on research design is available in the Nature Portfolio Reporting Summary linked to this article.

## Data availability
All data supporting the findings of this study are available within the article and its supplementary files. Any additional requests for information can be directed to, and will be fulfilled by, the corresponding authors. Source data are provided with this paper.

## Code availability

Custom code developed by CytoTronics Inc. was used to interface with the electronic plate, run the measurements, and collect the data. A custom data pipeline developed by CytoTronics Inc. was used to output individual and aggregate pixel values from the raw electronic data. All custom code is proprietary to commercial products and is not publicly available.

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

## Acknowledgements

The authors are grateful for the support of this research by Samsung Advanced Institute of Technology, Samsung Electronics, Suwon, Republic of Korea (A37734 to D.H. and A37738 to H.P.), the Army Research Office (W911NF-15-1-0565 to D.H.), the Army Research Office (W911NF-17-1-0425 to D.H.), and the Gordon and Betty Moore Foundation (to H.P.). M.B. was supported by Maximizing Investigators' Research Award (5R35GM137895) and the Harvard Medical School Junior Faculty Armenise grant. Select figures (specifically Fig. 1d, Supplementary Fig. 4, and Supplementary Fig. 5 were created using Biorender.com).

## Author contributions

J.A., H.P., D.H., M.B., D.S., A.M., W.W., and S.C. conceived and designed the experiments. W.W., H.H., and J.A. designed the CMOS integrated circuit, W.W., M.C., Z.B., A.A., and J.A. designed the electronic platform, H.M., B.P., and H.W. designed the software environment, S.C., W.W., A.M., H.L., P.S., I.N., C.A., R.S.G., and J.A. performed the experiments, P.S., H.W., and J.A. performed data analysis, M.B., D.H., H.P., and J.A. supervised the collaborative project. S.C. and J.A. wrote the manuscript, and all authors read and discussed it.

## Competing interests

W.W. and J.A. contributed to the work at Harvard University and transitioned to employment at CytoTronics. S.C., H.L., P.S., I.N., C.A., H.M., B.P., H.W., M.C., Z.B., A.A., and D.S. were employed at CytoTronics during their contributions to the work. D.H. and H.P. hold shares in CytoTronics. D.H. is also a Fellow of Samsung Electronics, but there is no competing interest for this work D.H. performed exclusively at Harvard University. The other authors declare no competing interests.
