## [Peer review file · Nature Communications]

REVIEWER COMMENTS

Reviewer #1 (Remarks to the Author):

The authors demonstrate a high throughput system based on custom CMOS chips that are packaged into a 96 well plate format. Although much of the interfacing is implemented through a PCB and the lab-on-CMOS is not original, it is a significant first step demonstration in combining lab-on-CMOS with the traditional well plate format used by biologists. The time to scan a plate is very good. It would be good to understand how this would increase if the arrays is scaled up even more. Impedance measurement is very similar to capacitive based Lab-on-CMOS sensors (e.g. Smith K, Lin CY, Gilpin Y, Wayne E, Dandin M. Measuring and modeling macrophage proliferation in a lab-on-CMOS capacitance sensing microsystem. *Front Bioeng Biotechnol.* 2023 May 12;11:1159004. doi: 10.3389/fbioe.2023.1159004. PMID: 37251577; PMCID: PMC10213696.). A comment on the comparison of the techniques could be helpful.

The experiments demonstrate that impedance can be measured. From the data, the different frequencies allows different information to be determined. This however leads to the question of why not just do a scan of all frequencies? Obviously this will increase the measurement time, but the tradeoffs involved are not clear. Overall a good well written paper.

Reviewer #2 (Remarks to the Author):

This manuscript by Chitale et al. describes a novel system for phenotypic cell characterization, which adds the benefits of spatial and temporal resolution of near single cell detail in live cells. The authors describe a system that is based on cell impedance to measure changes in cell adherence, motility, among other parameters, in a semi-high throughput manner. The manuscript is well written, the description of the technology is straightforward and the methodology represents a great improvement compared to current label-free technologies. A few suggestions are listed that if addressed would further heighten the excitement for this manuscript.

1) While the methodology offers impressive detail in assessing cellular functions and morphology, it would be valuable to make direct comparisons with established methods. The authors make claims about the superiority of their system, however without direct comparisons to other, impedance-based methods, it is up to the reader to either accept or reject their claims. This is particularly important in Figs. 3 and 4, which compares several different cell lines. It would be useful to demonstrate at least some data using a more established method, to demonstrate that this novel technique will replicate results obtained with different cell types.

2) Similarly, in the compound screen (summarized in Fig 6), the authors make claims regarding the MOA of screening compounds based on their co-clustering with compounds of known bioactivity. This data would be strengthened by the addition of confirmatory results using an orthogonal method, e.g. direct measurement of DNA damage or proteasome inhibition.

3) Finally, while the technology described here offers an advancement compared to currently available methods, some discussion regarding plans for further miniaturization and/or increasing throughput would be informative. The authors claim that their methodology is 'high-throughput', however this term seems somewhat overstated at this point. Rather, 'a technology with the potential for HTS' would be more appropriate, with the added discussion on the feasibility of screening 10,000s of compounds.

Point-by-Point Responses to the Reviewers' Comments

Response to Reviewer #1

The authors demonstrate a high throughput system based on custom CMOS chips that are packaged into a 96 well plate format. Although much of the interfacing is implemented through a PCB and the lab-on-CMOS is not original, it is a significant first step demonstration in combining lab-on-CMOS with the traditional well plate format used by biologists.

We thank Reviewer #1 for his/her positive remarks on our CMOS technology integration into the standard wellplate format.

The time to scan a plate is very good. It would be good to understand how this would increase if the arrays is scaled up even more.

We appreciate Reviewer #1's question about scan time and electrode array scaling. In response, we have added additional discussion on page 8 (bolded below) around the compromise between number of electrodes/pixels, frequency, and measurement time including a new reference [Hedayatipour et al., "CMOS based whole cell impedance sensing: Challenges and future outlook." *Biosensors and Bioelectronics* 143 (2019): 111600].

“The total scan time is a function of the lowest measured frequency (at least 4 ms per electrode for 250 Hz), total electrodes and number measured simultaneously (4,096 electrodes total with 16 measured at a time), configuration programming time, and number of wells measured simultaneously (6 wells total)¹⁶. Each field geometry takes 40 s to scan the full plate; the 3 fields are typically performed in sequence resulting in 120 s (2 min) for a full scan of the 27 impedance parameters. For most cell types and experiments, we found that performing full scans every 15 minutes balances total data size (~17 GByte/72 hour experiment) and time-course resolution. For scaling arrays to more electrodes while maintaining a similar scan time, the lowest frequency measured could be increased, or improved electronics could be implemented to scan more electrodes and/or wells simultaneously¹⁶.”

*Impedance measurement is very similar to capacitive based Lab-on-CMOS sensors (e.g. Smith K, Lin CY, Gilpin Y, Wayne E, Dandin M. Measuring and modeling macrophage proliferation in a lab-on-CMOS capacitance sensing microsystem. *Front Bioeng Biotechnol.* 2023 May 12;11:1159004. Doi: 10.3389/fbioe.2023.1159004. PMID: 37251577; PMCID: PMC10213696.). A comment on the comparison of the techniques could be helpful.*

We thank Reviewer #1 for bringing this capacitive based work to our attention. We've added a comparison sentence referencing the work and another state-of-the-art capacitive work [Hu et al., "Super-resolution electrochemical impedance imaging with a 512× 256 CMOS sensor array." *IEEE Transactions on Biomedical Circuits and Systems* 16, no. 4 (2022): 502-510] to compare to our technique, added on page 7:

“In comparison to other works which measure high frequency (>1 MHz) cross electrode capacitance¹⁷ or paired electrode capacitance changes¹⁸, our field-based impedance measurements distinguish multiple low-frequency tissue and cell parameters far away from the Debye capacitance sensing region, as discussed below. Higher frequency field measurements could accomplish similar capacitive sensing^{17,18} to reveal different cell

properties^{8,16}, but would require higher bandwidth op-amps at the tradeoff of more power consumption.”

The experiments demonstrate that impedance can be measured. From the data, the different frequencies allows different information to be determined. This however leads to the question of why not just do a scan of all frequencies? Obviously this will increase the measurement time, but the tradeoffs involved are not clear.

We greatly appreciate Reviewer #1’s comment and question on scan time and measurement frequency tradeoffs. We’ve addressed this question by expanding our measurement technique explanation and adding a new reference [Hedayatipour et al., "CMOS based whole cell impedance sensing: Challenges and future outlook." Biosensors and Bioelectronics 143 (2019): 111600] on page 7,

“Therefore, for each field measurement, four frequency signals are digitally added together and applied to the active electrode(s). A Fast-Fourier Transform (FFT) then calculates the four frequency magnitudes and phases and a Direct Current magnitude to create 9 impedance parameter maps – 27 total maps when measuring all 3 field configurations (Methods). The simultaneous multi-frequency approach is faster than sweeping the frequency to reduce scan time¹⁶ and the upper frequency (16 kHz) is limited by our amplifier bandwidth.”

and on page 8,

“The total scan time is a function of the lowest measured frequency (at least 4 ms per electrode for 250 Hz), total electrodes and number measured simultaneously (4,096 electrodes total with 16 measured at a time), configuration programming time, and number of wells measured simultaneously (6 wells total).”

Overall a good well written paper.

We thank Reviewer #1 for his/her positive remark, and the overall helpful comments and suggestions.

Response to Reviewer #2

This manuscript by Chitale et al. describes a novel system for phenotypic cell characterization, which adds the benefits of spatial and temporal resolution of near single cell detail in live cells. The authors describe a system that is based on cell impedance to measure changes in cell adherence, motility, among other parameters, in a semi-high throughput manner. The manuscript is well written, the description of the technology is straightforward and the methodology represents a great improvement compared to current label-free technologies.

We thank Reviewer #2 for his/her positive comments on the manuscript.

A few suggestions are listed that if addressed would further heighten the excitement for this manuscript.

We greatly appreciate Reviewer #2’s suggestions to increase the impact of the manuscript.

1) While the methodology offers impressive detail in assessing cellular functions and morphology, it would be valuable to make direct comparisons with established methods. The authors make claims about the

superiority of their system, however without direct comparisons to other, impedance-based methods, it is up to the reader to either accept or reject their claims. This is particularly important in Figs. 3 and 4, which compares several different cell lines. It would be useful to demonstrate at least some data using a more established method, to demonstrate that this novel technique will replicate results obtained with different cell types.

We appreciate Reviewer #2's request for comparison to other impedance techniques. In general, our measurements quantify multiple functional aspects of cells. While other instruments (both impedance based and non-impedance based) measure some of these features, to our knowledge there is no method that combines our readouts into one platform for a direct comparison. To address Reviewer #1's comment, we added comparisons to other impedance devices on the market which measure similar types of cells and made additional discussions (bolded below). As part of these edits, we've added 5 references of papers performing impedance measurements of multiple types of cells. The edits are made on pages 12-13,

“To this end, other impedance methods have also reported differences between cell types based on impedance readouts³²⁻³⁶. For example, our observation that Calu-3 cells have a higher barrier than A549 cells was previously observed using a multi-well impedance device (Axion Biosystems)³³ and by conventional trans-well TEER measurements³⁶. However, we additionally observe that A549 cells have a higher attachment and motility than Calu-3 cells, features that other impedance techniques cannot observe. Of note, the normalized barrier resistance taking into consideration the effective unit area of each electrode [$25 \times 25 \mu\text{m}^2$] for Calu-3 ($\sim 200 \Omega \cdot \text{cm}^2$) and A549 ($\sim 20 \Omega \cdot \text{cm}^2$) match value ranges from various literature sources using traditional TEER readouts³⁶, thus validating our measurement method.”

page 14 and 15,

“MCF-7 shows an epithelial morphology and expression of E-cadherin while MDA-MB-231 shows a mesenchymal phenotype with no expression of E-Cadherin²⁹. **Of note, currently used impedance devices have been unable to distinguish between these cell types in head-to-head comparisons³².** We plated MCF7 and MDA-MB-231 cells at a low-plating density (10,000 total cells per well) and different ratios to measure differences in growth characteristics over 72 hours (Fig. 4 and Methods)...

...Leveraging spatial information to assess population statistics over time can reveal subtle responses of culture sub-populations to stimuli and are more sensitive than cumulative, aggregate well readout approaches. **For example, though previous literature could assess the ‘energy’ of a culture to compare cancerous versus non-cancerous cell types³², they are not able to perform population statistics within the well itself to assess heterogeneity as there is no spatial information.”**

and on page 20 and 21 in the discussion,

“Currently, CMOS devices are used mostly for neural applications, while most macroscale electrode impedance devices are only able to measure a bulk impedance signal per well and lack the ability to measure orthogonal parameters in diverse cell types. Multiple biological processes such as cell death, loss of barrier, and change in attachment of cells are all reflected in the change in impedance per well and cannot be deconvolved. By combining field geometries and frequencies, we can measure distinct independent properties such as cell death, barrier, attachment, flatness, and motility.”

2) Similarly, in the compound screen (summarized in Fig 6), the authors make claims regarding the MOA of screening compounds based on their co-clustering with compounds of known bioactivity. This data

would be strengthened by the addition of confirmatory results using an orthogonal method, e.g. direct measurement of DNA damage or proteasome inhibition.

We thank Reviewer #2 for this detailed observation. In a traditional screening approach, one would attempt to elucidate the bioactivity of unknown compounds. However, in our proof-of-concept screen, all the compounds screened are annotated with well-characterized bioactivity. Thus, we were able to retroactively use the known bioactivity to examine functions that clustered together based on their effect on cells. We highlight a few examples and provide the list of compounds within the clusters in the table below. It is important to note that compounds with different functions can cluster together due to having similar effects on cell state/morphology. When used for drug discovery, we imagine that a set of well-defined positive controls will be essential to identify compounds that cause an effect of interest.

Cluster #1 is largely comprised of compounds that cause G2/M arrest, primarily through perturbation of spindles. Thus, these compounds lead to a loss of confluence and altered cell geometry (increased cell flatness) as measured by our platform (Fig. 6d-e). Cluster #3 is largely comprised of compounds that induce or inhibit the repair of double-stranded breaks (DSBs). DSBs are one of the most lethal types of DNA lesions leading to apoptosis and cell-death. Our platform measures a drastic drop in cell confluence and a loss in cell surface attachment during cell death. Cluster #5 is comprised of CDK inhibitors and DNA damage-inducing compounds – both of which cause cell-cycle arrest, however with a very different mechanism than microtubule inhibition. Interestingly, these compounds have a more subtle effect on cell confluence and lead to an increase in attachment.

Cluster	Compound	Target: function
Cluster 1 (Cytoskeletal)	HMN-214	PLK: causes mitotic arrest
	Vincristine sulfate	Microtubule: causes mitotic arrest, inhibits microtubule polymerization
	Combretastatin A4	Microtubule: causes mitotic arrest, inhibits tubulin polymerization
	Picropodophyllin (PPP)	IGF-1R: mitotic arrest, depolymerizes microtubules
	Monomethyl auristatin E (MMAE)	Microtubules: disrupts microtubule networks
	Irinotecan HCl Trihydrate	Topoisomerase: induces spindle damage, cell cycle arrest
	Pixantrone Maleate	Topoisomerase: cell cycle arrest
	3-Nitrocoumarin	Phospholipase C-g: disrupts actin-dependent tight junctions
Cluster 3 (DNA damage - double stranded breaks)	Gemcitabine	DNA/RNA Synthesis: inhibits DSB repair
	SRT 1720 (Hydrochloride)	SIRT: disrupts recognition of DSBs
	AZ6102	PARP: inhibits DNA resection at DSBs
	Daunorubicin HCl	Topoisomerase : induces DSBs
	IMD-0354	IKK: inhibits repair of DSBs
	Izorlisib	PI3K: inhibits the key proteins involved in DSB repair
	GSK-3 inhibitor 1	GSK-3: disrupts DSB repair
	Branaplam	DNA/RNA Synthesis
	TAK-220	CCR; HIV
	Gemcitabine	DNA/RNA Synthesis: inhibits DSB repair
	SRT 1720 (Hydrochloride)	SIRT: disrupts recognition of DSBs
Cluster 5 (DNA damage / CDK inhibitors)	Dynasore	Dynamain: prevents endocytosis, G0/G1 cell cycle arrest
	BF738735	PI4K: DNA damage, which induces cell cycle arrest
	Raltitrexed	Thymidylate Synthase: DNA damage, G0/G1 cell cycle arrest
	NSC 625987	CDK: inhibits cell proliferation
	MC180295	CDK: inhibits cell proliferation
	CDK5 inhibitor 20-223	CDK: inhibits cell proliferation
	BSJ-4-116	CDK; PROTACs: cell cycle arrest

Cluster	Compound	Target: function
	CI-1044	Phosphodiesterase (PDE): induces apoptosis and cell cycle arrest
	AZD1656	Glucokinase

We have incorporated these points in the manuscript's discussion of the screen starting on page 18 and 19,

“All the compounds screened are annotated with well-characterized bioactivity. Thus, we were able to retroactively use the known bioactivity to examine functions that clustered together based on their effects on cells. In some cases, compounds with the same annotated MOA cluster together, pointing to similar effect on cell state. However, compounds with different MOA may also cluster together, based on their effect on related functional/morphological pathways. To gain deeper insights into the separation of compounds into specific clusters and their effects on cell state, we examined time traces for selected clusters (Fig. 6d). The traces showed orthogonal measurements across the morphological parameters, differing in both magnitude and behavior over time. To facilitate comparison, we summarized the time traces using quantitative metrics termed bio-basis (Fig. 6e, details in Methods). Radar plots revealed the strength of our technology in classifying compounds causing cell death. Of the 6 clusters highlighted, 4 consist of compounds that affect cell proliferation or cause cell death through inhibiting DNA synthesis or cell cycle (**clusters 1, 2, 3 and 5**). **In addition to the effects on cell confluence, each cluster exhibited distinct morphological effects and therefore varied bio-basis specific to their respective MOA. Compounds targeting microtubules (cytoskeletal signaling) or causing G2/M arrest (cluster 1) affect cell shape as measured by flatness. DNA synthesis inhibitors (nucleotide analogues) (cluster 2) primarily affect cell attachment and dynamicity. Compounds that either induce or inhibit the repair of double stranded DNA breaks (cluster 3), one of the most lethal types of DNA lesions, produce the highest cell death rate and a significant loss in attachment during cell death. CDK inhibitors (cluster 5) affect barrier strength, cell flatness and staticity (reduced cell movement). Cluster 6, comprised of various kinase inhibitors, causes a rapid and drastic loss in barrier, attachment, and movement.** These functional insights provide new information on effects of known compounds and can help link unknown compounds to specific cellular pathways. Additionally, we identified a cluster primarily composed of GPCRs involved in neuronal signaling (cluster 4), which displayed a subtle yet rapid detachment response. This type of transient response can only be captured by our platform's high spatial and temporal resolution across parameters.”

3) Finally, while the technology described here offers an advancement compared to currently available methods, some discussion regarding plans for further miniaturization and/or increasing throughput would be informative. The authors claim that their methodology is 'high-throughput', however this term seems somewhat overstated at this point. Rather, 'a technology with the potential for HTS' would be more appropriate, with the added discussion on the feasibility of screening 10,000s of compounds.

We appreciate Reviewer #2's observation. Approaching the problem from the lens of single-well CMOS devices, we view our multi-well and multi-plate platform as high-throughput but appreciate that other techniques can achieve much higher screening scales. As the measurement systems are integrated into the plate, in theory our platform is scalable by providing a larger number of USB connectors with a corresponding number of incubators and other infrastructure. In fact, our company is pursuing this exact line of development. To reflect Reviewer #2's comment, we've edited (bolded below) our opening discussion statement on page 20 to qualify our throughput claims,

“In this work, we describe the design and implementation of a custom semiconductor 96-microplate device **with the potential to scale to a high-throughput screening system.**”

and added additional discussion around scaling to high-throughput, also on page 20,

"Scalability and versatility are crucial attributes for a technology employed in drug discovery. Our miniaturized data acquisition system allows parallel operation of multiple plates directly in the incubator, enabling unlimited scalability. In contrast, other impedance devices utilize an “instrument” design where plates plug into a box for measurement, reducing scalability. **For eventual utility in drug discovery where 10,000s of compounds are routinely screened, additional miniaturization of our 96-microplate to more dense form factors (e.g., 384 wells per plate) and/or more plates per incubator setup will be required. Major considerations to achieve this scale include management of high-speed data interfaces, thermal considerations with electronic power, and data processing pipelines.**”

REVIEWERS' COMMENTS

Reviewer #1 (Remarks to the Author):

The authors have addressed most of the comments. Some minor comments remain. While the comparison with a few other systems was added, the authors should consider to add a more comprehensive table.

Authors should consider adding references for the "well known bioactivity" as the paper seems targeted towards a potentially interdisciplinary audience, not all readers will agree that this is well known (hence the original comment on making orthogonal measurements).

Reviewer #2 (Remarks to the Author):

In their revised manuscript, Chitale et al. have adequately responded to previous reviewer comments and the majority of the previously raised concerns have been addressed. In response to previous comments, the authors include a table of clustered compounds based on MOA, that support their observations using their method. It would strengthen the manuscript if this table was included as supplemental data in the manuscript.

Point-by-Point Responses to the Reviewers' Comments

Response to Reviewer #1

The authors have addressed most of the comments. Some minor comments remain. While the comparison with a few other systems was added, the authors should consider to add a more comprehensive table.

We thank the Reviewer #1 for his/her positive feedback. We have added a table to directly compare our method to other impedance and CMOS based methods. The table is now included in the manuscript as Supplementary Table 1.

Authors should consider adding references for the "well known bioactivity" as the paper seems targeted towards a potentially interdisciplinary audience, not all readers will agree that this is well known (hence the original comment on making orthogonal measurements).

We thank the Reviewer #1 for this suggestion. We have added in details of the libraries used for high-throughput screening in the Methods, to justify our claim of the compounds having well-known activity. In addition, all the compounds included in the study include citations to previous studies characterizing their bioactivity.

Response to Reviewer #2

In their revised manuscript, Chitale et al. have adequately responded to previous reviewer comments and the majority of the previously raised concerns have been addressed. In response to previous comments, the authors include a table of clustered compounds based on MOA, that support their observations using their method. It would strengthen the manuscript if this table was included as supplemental data in the manuscript.

We thank the Reviewer #2 for his/her positive comments. We incorporate the table in the manuscript as Supplemental Table 2.